# Hyper-HMM: aligning human brains and semantic features in a common latent event space

**Caroline S. Lee**
Department of Psychology
Columbia University
New York, NY
cl4353@columbia.edu

**Jane Han**
Department of
Psychological and Brain Sciences
Dartmouth College
Hanover, NH
Jane.Han.GR@dartmouth.edu

**Ma Feilong**
Center for Cognitive Neuroscience
Dartmouth College
Hanover, NH
Feilong.Ma@dartmouth.edu

**Guo Jiahui**
Center for Cognitive Neuroscience
Dartmouth College
Hanover, NH
Jiahui.Guo@dartmouth.edu

**James V. Haxby**
Center for Cognitive Neuroscience
Dartmouth College
Hanover, NH
james.v.haxby@dartmouth.edu

**Christopher Baldassano**
Department of Psychology
Columbia University
New York, NY
c.baldassano@columbia.edu

## Abstract

Naturalistic stimuli evoke complex neural responses with spatial and temporal properties that differ across individuals. Current alignment methods focus on either spatial hyperalignment (assuming exact temporal correspondence) or temporal alignment (assuming exact spatial correspondence). Here, we propose a hybrid model, the Hyper-HMM, that simultaneously aligns both temporal and spatial features across brains. The model learns to linearly project voxels to a reduced-dimension latent space, in which timecourses are segmented into corresponding temporal events. This approach allows tracking of each individual's mental trajectory through an event sequence and for alignment with other feature spaces such as stimulus content. Using an fMRI dataset in which students watch videos of class lectures, we demonstrate that the Hyper-HMM can be used to map all participants and the semantic content of the videos into a common low-dimensional space, and that these mappings generalize to held-out data. Our model provides a new window into individual cognitive dynamics evoked by complex naturalistic stimuli.

## 1 Introduction

Dynamic, continuous stimuli such as movies, stories, music, or educational videos evoke complex response patterns throughout the brain that can be captured using functional magnetic resonance imaging (fMRI). In addition to cognitive processes shared by all participants, these responses reflect individual differences in perception, learning, and memory. These differences can result in the same semantic concept being represented at varying points in time or by different spatial patterns of brain activity. An ongoing challenge is to preserve these heterogeneous dynamics across individuals while capturing shared cognitive processes underlying human intelligence.

37th Conference on Neural Information Processing Systems (NeurIPS 2023).

One line of research has focused on the *temporal* differences between participants. Previous work using a Hidden Markov Model (HMM) approach [1] has found that there are differences in the timing of event representations across development [9] or from repeated exposure to a stimulus [27], and has measured relationships between event timing and stimulus interpretation in individuals [41]. This approach requires either making the assumption that the event-specific spatial activity pattern across voxels is identical across participants, or fitting completely separate models for each participant (preventing the identification of corresponding events across people).

A separate category of approaches has focused on the *spatial* differences across participants, learning a functional alignment across participants that accounts for differences in functional architecture [6, 16, 19]. However, these approaches assume that all participants' neural responses are exactly synchronized in time. This assumption may be strongly violated, especially when studying idiosyncratic cognitive processes such as learning complex educational material.

Another challenge when using naturalistic stimuli is identifying the stimulus features that are driving neural activity. A common approach is to construct a vector representation of a particular class of features in the stimulus, and then construct a linear "encoding" or "decoding" model that maps these vectors onto patterns of brain activity or vice versa [21, 34, 47]. However, these models again face the same challenges of temporal and spatial alignment; they assume that the neural response to a stimulus is immediate (or occurs at a short, fixed delay learned by the encoding model) and that all participants represent stimulus features with the same pattern of brain activity (or are restricted to modeling a single participant at a time).

**Contributions.**    Here we propose the Hyper-HMM (H-HMM), an extension of the HMM from Baldassano et al. [1] that can simultaneously capture temporal *and* spatial differences across participants. The model can also align across entirely different kinds of feature spaces, creating mappings between neural data and a semantic embedding of the stimulus. We validate our model using simulated data and by using fMRI data from individuals watching course videos for a Computer Science class [30]. We can successfully map both the human participants and an embedding model of stimulus into a common latent space of semantic events, and this mapping generalizes to held-out videos in this dataset. Our novel approach is broadly applicable to any neuroimaging dataset with continuous naturalistic stimuli, and can provide a new way of characterizing and studying individual differences in cognitive processes.

## 2   Background

### 2.1   Temporal alignment between brains and with stimuli

Although neural responses are assumed to be tightly temporally-locked to the stimulus during traditional experimental designs with discrete stimuli, this is often not the case for more abstract cognitive trajectories evoked in naturalistic experiments. Especially in high-level regions, we could see responses that lag behind the stimulus (if a viewer takes time to comprehend an event), run ahead of the stimulus (if a viewer can predict upcoming events [27]), or are generated based on a past stimulus rather than an ongoing stimulus (such as during narrative recall [5]). As a further complication, these temporal shifts could differ in meaningful ways across people [41]. One approach for identifying a temporal alignment across datasets is to use Dynamic Time Warping, which stretches and compresses the temporal axis of each dataset to maximize their alignment [44]. This technique makes relatively few assumptions about the structure of the temporal alignment, but does require that the datasets can be matched together at the level of individual timepoints (i.e. that every 1-2 seconds of the stimulus can be directly mapped to 1-2 seconds of the neural response).

An alternative approach is to assume that datasets can be segmented into discrete, cognitively-meaningful events on the timescale of tens of seconds to minutes. The theory that people naturally perceive and remember continuous stimuli as individual events has been extensively studied in cognitive psychology [39, 50], and signatures of this event segmentation can also be measured using neuroimaging [1, 49]. This work therefore suggests that events are a natural unit for representing cognitive dynamics and for performing temporal alignment, and we can employ Hidden Markov Models (HMMs) to track these discrete cognitive states [1, 27, 45]. Current HMMs for neuroimaging, however, assume that the neural signature of an event is a specific spatial pattern of activation across voxels in a region of interest (ROI) which is identical across all experimental subjects.

## 2.2 Spatial hyperalignment between brains

Since the early days of fMRI, researchers have recognized that the specific spatial arrangement of functional units in the brain differs across individuals, motivating the use of "localizer" tasks to identify the functional layout of a specific brain [42]. One kind of method for performing multi-subject analyses is to use responses to a long naturalistic stimulus to learn a linear transformation that maps between individuals, a method called "hyperalignment" [18]. There are now many variations and applications of this approach [19, 22], including models that map subjects into a shared lower-dimensional space [4, 6, 11, 16]. Though these methods can capture spatial variation in functional responses, they assume perfect temporal alignment across participants in their neural responses.

## 2.3 Mapping between stimulus features and neural responses

Extensive work has focused on bidirectionally mapping between features of the external world and the human brain [34, 36]. These can take the form of encoding models that predict brain activity patterns as a function of stimulus features [21, 31, 33, 43, 47], decoding models that produce estimates of external stimuli (or internal mental states) given brain activity [20, 32, 35], or hybrid models that capture structure among both stimulus features and brain activity [23]. These models assume that the temporal correspondence between the stimulus and neural responses is known, or allow for a small fixed lag in the neural responses by including time-shifted versions of the stimulus features.

# 3 Methods overview

## 3.1 Model description and fitting

The Hyper-HMM (Figure 1) combines the strengths of these existing methods, removing the strong assumptions of spatial or temporal alignment across individuals and temporal synchronization to the stimulus. Instead, we require only a much more abstract level of correspondence across datasets, assuming that: a) all subjects and stimulus models proceed through the same sequence of semantic states (events); b) each event $e$ can be represented as a low-dimensional vector $G_e \in \mathbb{R}^D$ (where the latent dimensionality $D$ is a parameter of the model); c) each subject and model can be mapped into this latent space using a linear transformation $W_i$ (unique to each subject and model); d) in the latent space, all timepoints during which this subject or model was in event $e$ should be correlated with $G_e$.

To model both these temporal and spatial assumptions, we employ a chain-structured Hidden Markov Model in which all $n$ participants transition sequentially through a sequence of event states, as in Baldassano et al. [1]. For the observation model, we define

$$p(X_{i,t}|s_{i,t} = e) = \frac{1}{\sqrt{2\pi\sigma_m^2}} e^{-||z(W_i X_{i,t}) - z(G_e)||_2^2/(2\sigma_m^2)} \tag{1}$$

When a participant or stimulus model $i$ is in state (event) $s_{i,t} = e$, the latent-space mapping of the spatial voxel pattern or embedding vector ($W_i X_{i,t}$) should be similar to the group-level event representation $G_e$. All fMRI and stimulus datasets are therefore linked through their transform matrices $W_i$ into a common latent semantic space, which reflects between-event similarity structure shared by all fMRI subjects and with the stimulus features. The $z()$ function represents z-scoring (standardizing representations to have zero mean and unit variance), so that the Euclidean distance measure is proportional to the Pearson correlation between $W_i X_{i,t}$ and $G_e$.

We fit the model by iteratively estimating the temporal alignment $s_{i,t}$ for each dataset and then updating the shared event representations $G$ and the $W_i$ projection matrices. On each iteration, we project each subject's data and the stimulus features into the low-dimensional shared space using the current $W_i$ matrices. We apply the forward-backward algorithm from Baldassano et al. [1] in this shared space, probabilistically segmenting each timeseries into events corresponding to the patterns $G_e$. The resulting probabilities $\eta_{i,t,e} = p(s_{i,t} = e|X_i, W_i, G)$ can then be used to estimate the subject-specific voxel representations of each event as $E_{i,e} = \sum_t \eta_{i,t,e} X_{i,t} / \sum_t \eta_{i,t,e}$.

To update the group-level representations $G$, we stack the event patterns $E_i$ from all subjects and the stimulus (concatenating along the voxel/embedding dimension for each event, as in [16]) and then project down onto $D$ principal components. In order to ensure that the fMRI data and stimulus embedding have an equal vote in the shared event representations, the stimulus event patterns are

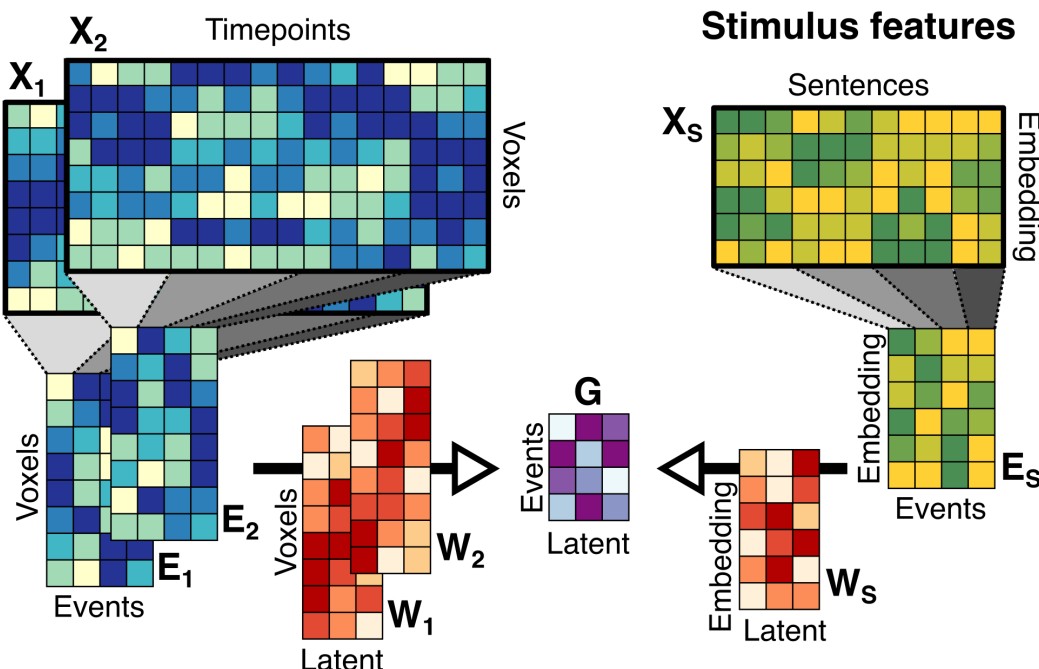

Figure 1: *Hyper-HMM alignment across multiple brains and stimulus features*. The H-HMM temporally divides each subject's brain data $X_i$ into discrete events with subject-specific patterns $E_i$, and temporally divides the stimulus embedding $X_S$ into event patterns $E_S$. The event patterns from all subjects and the stimulus are constrained to linearly project (through matrices $W_i$) to a shared, low-dimensional representation $G$.

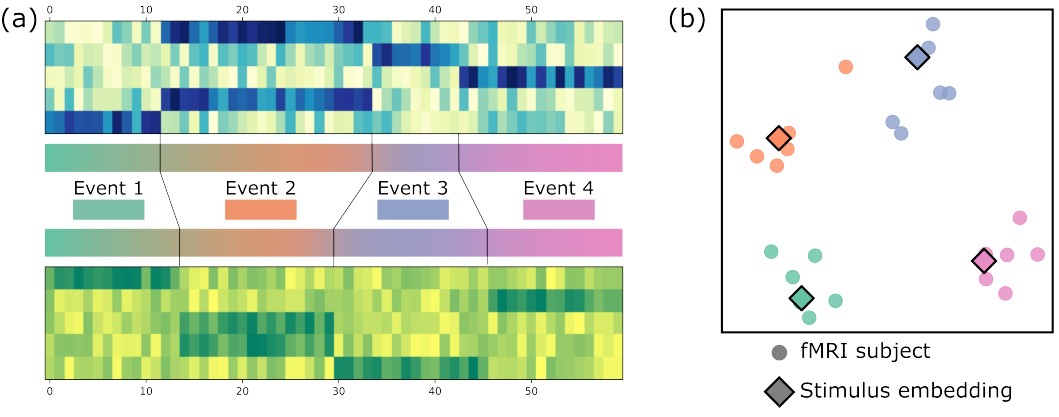

Figure 2: *Measuring learned event representations in the data*. a) The model is fit to voxel patterns (top) and stimulus (bottom) features, successfully learning a temporal alignment into corresponding events (middle) despite temporal and representational differences between the two datasets. b) Plotting the first two dimensions of the latent-space projection of event representations for fMRI and stimulus datasets shows clustering into corresponding events.

weighted $n$ times as strongly as each fMRI dataset. Finally, we update the transform matrices, $W_i$, using a ridge regression predicting $G$ from subject patterns (with $\alpha = 10$). We repeat this process (while annealing the HMM observation model variance $\sigma_m$ across iterations [1]) until the model's log-likelihood stops improving.

After fitting, the model provides a temporal segmentation into events that corresponds across subjects and between the subjects and the stimulus (Figure 2a). For each event, all subjects' spatial patterns of fMRI activity and the stimulus feature representation can be projected to the same region of latent space, through the learned matrices $W_i$ (Figure 2b).

## 3.2 Validation with simulated data

We tested the model's ability to recover ground-truth event structure under varying degrees of noise using simulated datasets. We used the fmrisim utility [13] in the BrainIAK toolbox [26] to generate a pure-noise dataset in the angular gyrus with the same number of subjects, runs, and timepoints per run as in the fMRI data used in our main experiments (see below). We simulated an event-structured signal by assigning blocks of timepoints to events, with events varying in length between 12 and 18 timepoints across subjects. Within each event there was a constant event pattern signal; these patterns came from the real fMRI data, averaged in 15-TR increments. By combining the signal and noise together with varying weights, we produced simulated datasets with realistic spatial and temporal correlation properties along with ground-truth timing and event patterns within each subject. We used these datasets to test the model's ability to recover subject-specific event boundaries and estimate subject-specific spatial projections that generalized across runs.

For each noise level, we trained the model on only one half of the runs, and then froze the projection matrices $W_i$ and performed temporal-only alignment on the held-out half of the runs. Ideally, these projection matrices should produce event representations on these held-out runs that are highly similar (across subjects and the stimulus embedding) for each event in the latent semantic space. To assess the degree of event clustering, we computed the sum of squared errors (SSE) between the latent-space event pattern for each fMRI subject and the mean pattern for this event across all subjects. We normalized this to yield a "variance explained" measure, dividing it by the SSE between all fMRI event patterns and the global mean pattern and then taking one minus this ratio. If the event patterns for all subjects map onto identical latent representations, this value will be 1, and patterns that show no clustering into events will yield a variance explained of 0. We repeated this process with the training and testing sets reversed, and averaged the results.

Next, we tested the model's ability to identify ground-truth event boundaries at each noise level. After fitting the model to the full dataset, we identified event boundaries for each subject as timepoints at which the most probable event switched, and computed the fraction of ground-truth boundaries that matched (within one timepoint) one of these model-derived boundaries.

## 3.3 Applying to an fMRI dataset

We used fMRI data collected by Meshulam et al. [30], in which undergraduate students ($n$=19) taking an introductory Computer Science course watched some of the course lecture videos in the scanner. There were five scan sessions throughout the course of the semester, divided into 21 separate scanning runs, for a total of 197 minutes of data per subject. We hypothesized that variations in expertise and concept understanding would lead to a relatively large amount of variability in neural responses across participants, making it ideal for testing our alignment approach. The fMRI data (TR = 2 seconds, voxel size = 3mm x 3mm x 3mm) was preprocessed using fMRIPrep [14], including mapping data to the cortical fsaverage6 surface, and then cleaned via linear regression to remove variance associated with motion, cerebrospinal fluid signals, white matter signals, and low-frequency drift. The timecourse of activity for each vertex was z-scored for each run.

We derived a semantic embedding of the lecture stimulus by transcribing the spoken dialogue using a speech recognition model [38] and then obtaining a vector representation of each sentence using a pre-trained transformer model [40]. Note that we do not make use of any timing information about the correspondence between fMRI timepoints and stimulus sentences, giving the H-HMM full flexibility in assigning stimulus timepoints to neural responses in individual subjects.

Here we focus on data from the angular gyrus and posterior medial cortex (as defined in Baldassano et al. [2]), given their role in anticipating familiar narrative stimuli and learning schematic structures [2, 27]. We fit the H-HMM to all 21 fMRI runs, setting the number of events such that the average event duration was 15 TRs (30 seconds). Each run had a separate set of group-level events $G$, but the $W_i$ matrices were constrained to be the same across all runs (i.e. the $W_i$ matrices were computed via a single regression from all subject/stimulus-specific events to all group-level events). For our primary analyses, we chose a latent dimensionality of $D = 3$ for the group-level shared semantic space (see Figure 7 for an exploration of alternative dimensionalities).

We examined the learned projection matrices $W_i$ in both neural and stimulus space. For fMRI data, we visualized the first dimension of $W_i$ (which captured the largest share of the variance in semantic space) by left-multiplying it by the Gram matrix $X_i X_i^T$ and then z-scoring. This produced a weighted average of the subject's spatial patterns, in which the weights were given by the similarity to the first dimension of $W_i$. We performed a similar operation for the stimulus embeddings, measuring the correlation between individual sentence embeddings and the first column of $X_S X_S^T W_S$. Sentences with particularly high or low correlations were those that were highly correlated or anticorrelated with this first semantic dimension.

To quantitatively evaluate these projection matrices, we used the variance-explained metric described in section 3.2, computing this value separately for 1) the clustering of the fMRI representations around their mean for each event and 2) the match between the stimulus representation and the mean fMRI representation for each event. As a baseline, we ran the same analysis, but used random projection matrices $W_i$ in which entries were sampled from a standard normal distribution $N(0, 1)$. We also repeated this analysis while varying the dimensionality $D$ of the latent semantic space.

**Compute resources.** Model fitting was performed using a maximum of 24 CPUs per fit. For fitting to the training data, compute time ranged from 12 - 16 minutes per split half, depending on the size of the ROI. The temporal-only alignment for the test data required 3 - 6 minutes for each fit.

## 4 Results

### 4.1 Simulated data

In our experiments with simulated fMRI data, the H-HMM can effectively learn spatial alignments across subjects even under relatively high levels of noise (Figure 3a). Additionally, the model can successfully recover the majority of the ground-truth event boundaries in each subject when the noise amplitude is up to twice as large as the signal amplitude (Figure 3b). These results demonstrate that the model and fitting procedure are robust to the noise properties of a typical fMRI dataset.

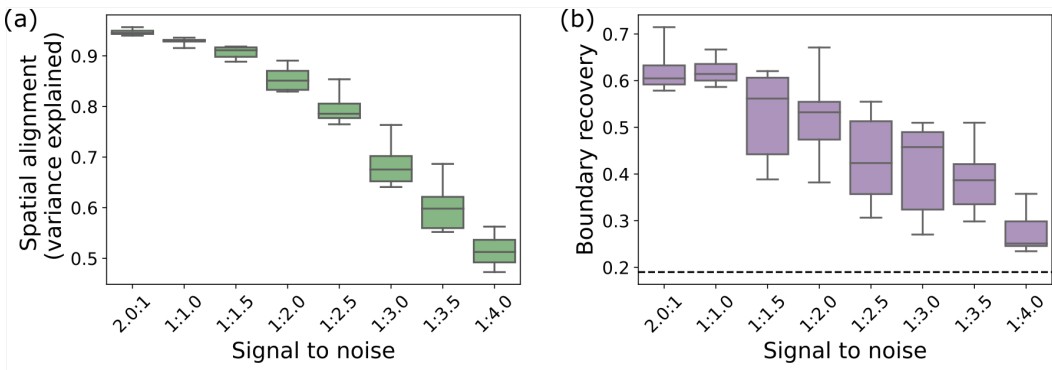

Figure 3: *Model validation in simulated data.* a) The model exhibits high alignment in the latent projections of held-out fMRI events (i.e. high variance explained by the event clustering) until the noise amplitude is several times larger than the simulated signal. b) The model can successfully recover ground-truth event boundaries for each subject up to high noise levels; the dotted line indicates the performance of a null model that randomly selects timepoints as event boundaries.

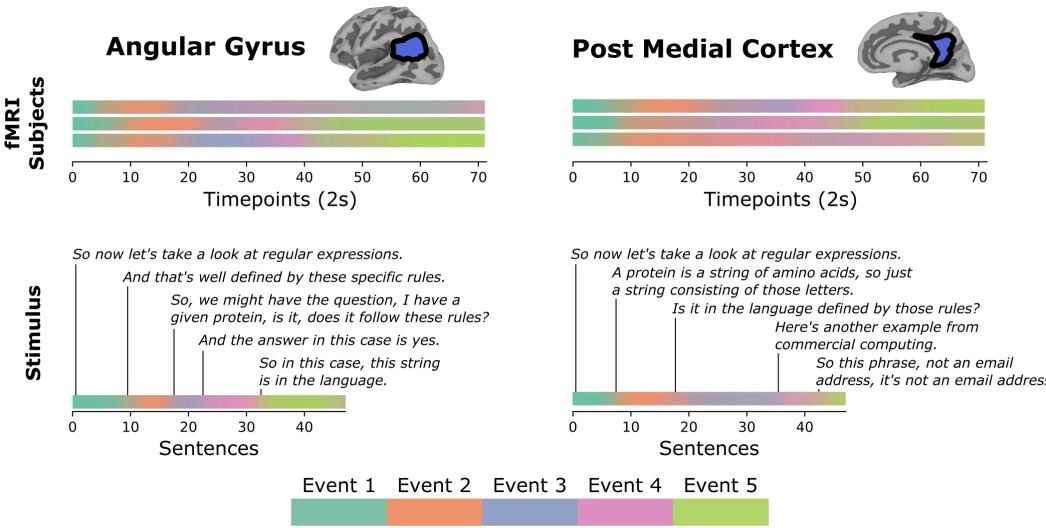

Figure 4: *Example of learned temporal alignment of fMRI subjects and stimulus features.* The H-HMM learns a segmentation of each neural timeseries and the stimulus embedding into corresponding events, with shared latent representations across brains and the stimulus. Here we visualize this temporal segmentation for the first half of one of the scan runs, for three of the subjects and the stimulus. The timing of transitions between cognitive states and the sharpness of these transitions vary across individuals. Sentences at the event transitions in the stimulus are shown, illustrating the semantic content present at the start of each event.

## 4.2 fMRI experiment

We fit the H-HMM to the full fMRI dataset for two brain regions (angular gyrus and posterior medial cortex) and examined the learned temporal and spatial alignment. An example of the temporal alignment for the first half of one run is illustrated in Figure 4, showing how each individual's neural responses and the stimulus sentences are segmented into a shared set of events. This kind of semantic event segmentation is made possible by the learned projections $W_i$ that align all subjects and the stimulus embedding into a common semantic space. A visualization of this alignment for the first (maximally-informative) semantic dimension is shown in Figure 5. Note that the overall spatial topography corresponding to this dimension is similar across subjects (e.g. in angular gyrus, the posterior portion has generally positive weights while the anterior portion has generally negative weights) but the fine-scale details are different in each subject, reflecting individual variation in how this semantic dimension is spatially expressed. We can also examine the learned projection of the sentence embeddings onto this semantic dimension. The poles of this first dimension in angular gyrus appear to correspond approximately to simple concrete sentences versus abstract statements, and in posterior medial cortex to future-oriented statements versus present-tense descriptions of program structure. These axes of semantic variation have been observed in prior fMRI studies: Conca et al. [10] show differences in activity for concrete vs abstract and future- vs past- and present-tense sentences, and concrete vs abstract concepts engage different regions of the semantic network [17].

In order to statistically verify that the learned projections to the latent space were meaningful, we next tested how well the projection matrices $W_i$ fit to one half of the data generalized to the other half of the data. We identified events in the testing data for both the neural and stimulus data, and measured the degree of clustering for corresponding events between fMRI subjects and between the fMRI group-average and the stimulus. An example of the event representations in the latent space is shown in Figure 6, illustrating that event patterns are projected into event-specific clusters. The event clustering for the fMRI subjects was substantially above a random-projection baseline in both regions ($p < 0.001$). The match between the neural representations and the stimulus representations was also significantly above baseline in both regions (Angular gyrus $p < 0.001$; PMC $p = 0.0027$). These results demonstrate that the learned mapping to a latent semantic space provides meaningful alignment between brains and provides a correspondence between brain data and stimulus features.

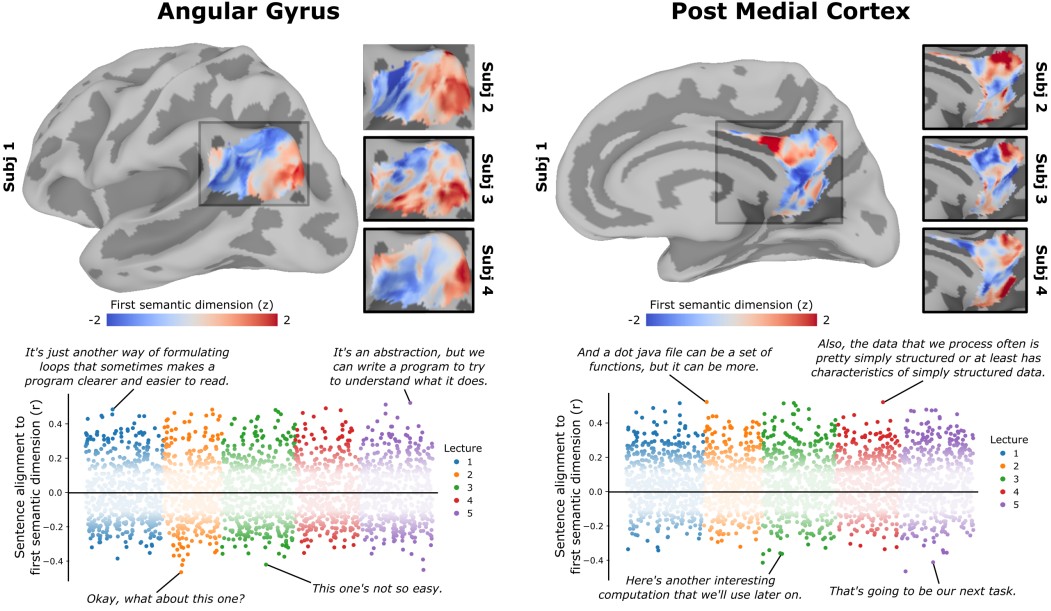

Figure 5: *Learned projection of voxels and stimulus features, for the first semantic dimension.* Fitting the H-HMM to the entire fMRI dataset produced projection matrices $W_i$ for each subject and for the stimulus embedding. The voxel patterns (for the first four subjects) and sentences associated with the maximally-informative learned semantic dimension are visualized for both the angular gyrus (left) and posterior medial cortex (right). The voxel maps are broadly similar across subjects, but have distinct fine-scale structures specific to each individual. The scatter plots show the association between each sentence and the first semantic dimension across the 2,445 sentences in the dataset, highlighting example sentences that are strongly correlated or anti-correlated with this dimension.

A key parameter of our model is the dimensionality $D$ of the latent semantic space. Intuitively, performance should suffer if $D$ is too small (preventing proper discrimination between semantically-different events) or if $D$ is too large (resulting in latent event representations that are largely orthogonal, which will fail to capture semantically-meaningful relationships and will fail to generalize to new events). We varied $D$ and again performed our analysis of held-out performance, fitting the model on half the data and measuring the degree of clustering on testing data here, including both fMRI and stimulus representations in our variance-explained clustering measure. We observed (Figure 7) that performance peaked for $D = 3$, with slightly reduced performance for $D = 2$ (the minimum dimensionality for which the HMM can compute pattern correlations) and substantially reduced performance as $D$ increased to $8$. We conclude that $D = 3$ provides the optimal number of semantic dimensions events in this dataset, though datasets that are very large or explore a broader set of semantic topics could require a higher-dimensional latent space.

## 5   Limitations, Future Work, and Conclusions

The Hyper-HMM makes several assumptions about the cognitive structure of naturalistic datasets that could be relaxed in future work. First, it assumes that all subjects and the stimulus proceed through the same linear sequence of cognitive states. This is a reasonable assumption for data collected during tasks such as watching or recalling movies with linear narratives [5]; but may be less accurate for tasks such as recalling a series of unrelated short videos [28]. Second, we assume that neural dynamics are relatively constant within cognitive states, with sharp jumps between states; this model would fail to capture dynamics in brain regions with brief responses occurring only at event boundaries, such as the hippocampus [3, 9]. We also assume that brain responses in different people as well as stimulus embeddings are all *linearly* projectable to a common semantic space. Future work could extend this projection to be non-linear, as has been explored for characterizing relationships between brain regions within individuals [15, 37].

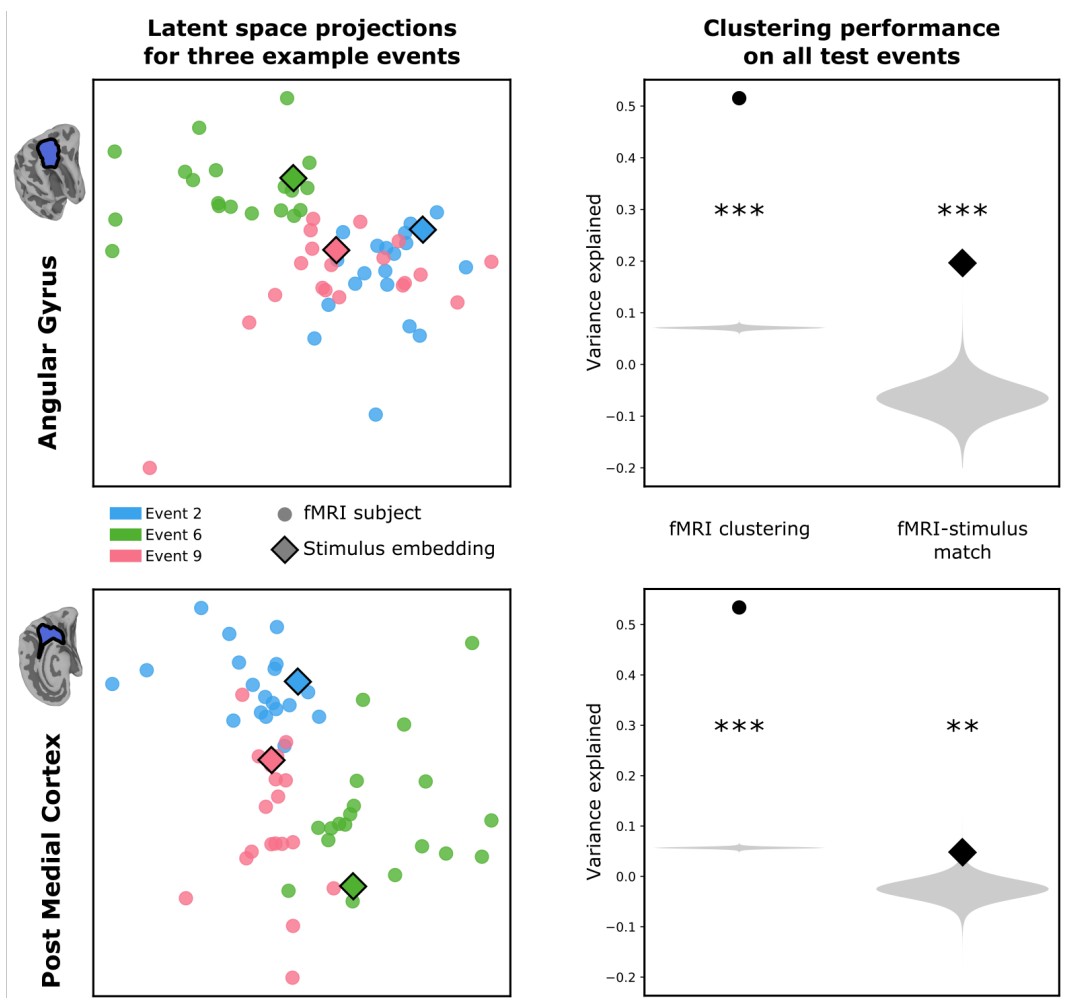

Figure 6: *Learned projections $W_i$ successfully generalize to held-out data.* The H-HMM was trained on half of the experimental runs, and then the $W_i$ matrices were fixed and temporal-only event alignment was performed on the held-out test data. The left panels show an example of these representations for three events in one held-out scanning run, showing that different events are projected to different parts of semantic space and that the stimulus events are well-aligned with the fMRI events. The right panels show the degree of clustering across all held-out events, measured separately for the clustering between fMRI subjects and for the match between fMRI data and the stimulus events. For both measures and both ROIs, clustering is significantly better than a random-projection baseline, indicating that our learned projection matrices generalize across experimental runs. ** $p < 0.01$, *** $p < 0.001$, computed as the fraction of null values greater than each result.

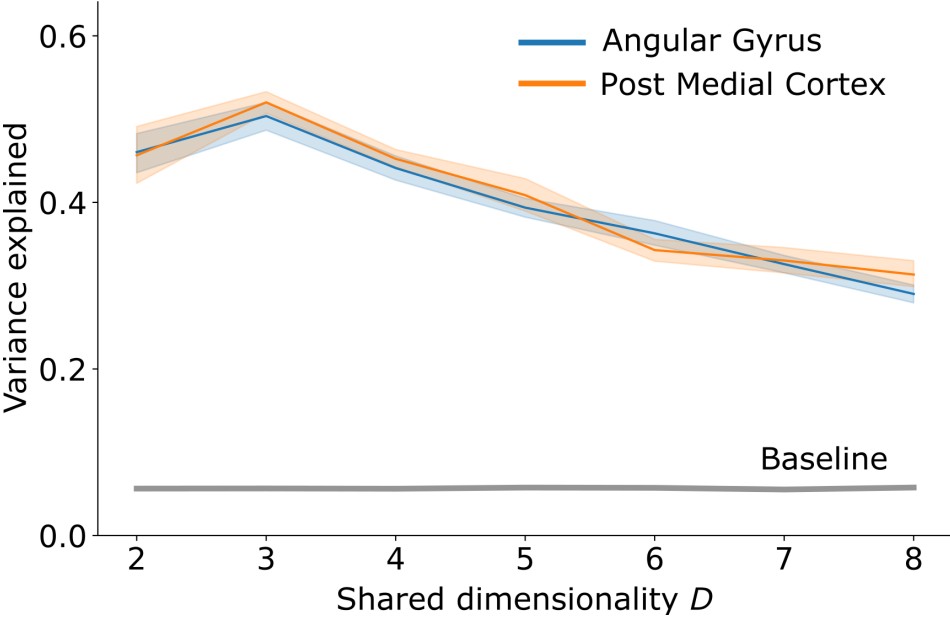

Figure 7: *Clustering performance when varying the dimensionality of the shared semantic space. Rather than using $D = 3$ as in all previous analyses, we tested the sensitivity of our results to the dimensionality of the shared latent space. For each choice of $D$, we computed the degree of clustering (among both fMRI events and stimulus events) in the latent space, and compared it to a random-projection baseline. We found above-baseline performance across a range of $D$ values, but clustering for held-out data is best for $D = 3$.*

Although our results and model-fitting have focused primarily on brain-stimulus alignment, our model can be expanded to alignment of cross-modal, or even cross-species, information in the brain. Current neuroimaging techniques each exhibit tradeoffs between spatial resolution and temporal resolution, and there have been several proposed approaches for combining information across experiments in which brain responses to the same stimulus were measured in different neuroimaging modalities [7, 8, 24]. The H-HMM could provide a new avenue for integrating the strengths of modalities with different spatial and temporal properties, even if measured in different groups of experimental subjects. Analogously, brain alignment across species, in particular between humans and primates, have provided insights into common representational processes [12, 25, 48], and the H-HMM could be applied to map cross-species representations into a common latent space.

The Hyper-HMM provides a highly-flexible framework for identifying and aligning event representations across people and to features of a stimulus. Our experiments demonstrate that fitting this model to an fMRI dataset is computationally and statistically feasible, and can learn meaningful mappings of brain responses and stimulus features into a stable semantic space. This model has broad applications to the rapidly-expanding field of studies using continuous naturalistic stimuli to study dynamic cognition [29, 46], including the characterization of individual differences in brain responses and the identification of representational properties of high-level brain regions.

## 6 Acknowledgements

We thank Meir Meshulam for discussions and guidance in incorporating his dataset in this project. In addition, we are grateful to the labs of Mariam Aly, Christopher Baldassano, and Jim Haxby for their feedback at multiple stages of presenting our analyses and ideas. Lastly, we thank the reviewers of this paper for their insights leading to improvements in our final version, especially in the design of the simulation experiments.

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

# Appendix

## Running the H-HMM code

Refer to README.md for a detailed explanation describing how to reproduce the main fMRI experiments, including the preprocessing steps for the fMRI data.

## Algorithm for fitting H-HMM

Algorithm 1 describes the H-HMM's fits to a single fMRI run. Fitting to all scan sessions is performed similarly, except that the forward-backward function is applied to each run separately. For full details see the README.md file in the attached code directory.

---

**Algorithm 1** HyperHMM algorithm

---

**Parameter:** $L$ = event length
**Parameter:** $D$ = latent dimension
$n$ = number of subjects
$T$ = number of timepoints
$V_i$ = number of voxels/features in subject/model $i$
$X_i$ = data for subject/model $i$ $(V_i \times T)$

$K \leftarrow \text{round}(T/L)$
$W_i \leftarrow N_{D \times V_i}(0,1) \; \forall i \in 1..n$
$G_e \leftarrow \text{mean}_i[\text{mean}_t[W_i X_i]] \; \forall e \in 1..K$
$\sigma \leftarrow 4$
**while** log-likelihood is improving **do**
  **for all** subjects $i$ **do**
    $\eta_i \leftarrow \text{forward\_backward}(\text{log\_probs}(W_i X_i, G, \sigma))$
    $E_{i,e} \leftarrow \sum_t \eta_{i,t,e} X_{i,t} / \sum_t \eta_{i,t,e} \; \; \forall e \in 1..K$
  **end for**
  $G \leftarrow \text{PCA}([E_i]_{1..n}, D)$
  $W_i \leftarrow [\text{ridge}(E_i, G)] \; \forall i \in 1..n$
  $\sigma \leftarrow \sigma * 0.98$
**end while**

---

