# OpenReview forum: "Hyper-HMM: aligning human brains and semantic features in a common latent event space"
_NeurIPS.cc/2023/Conference — NeurIPS 2023 poster_

### Official Review · Reviewer_XnCP · 2023-07-03

**Soundness:** 2 fair
**Presentation:** 2 fair
**Contribution:** 3 good
**Rating:** 4
**Confidence:** 4

**Summary:**

The authors propose an HMM-based model, Hyper-HMM, for characterizing variability in temporal and spatial dimensions in fMRI sequence datasets. The model is a chain-structured HMM where each discrete state (event) defines a relationship between neural activity and a stimulus embedding. Importantly, the discrete state defines the mean of the subject's neural activity projected to a lower-dimensional space. Each subject has a different lower-dimensional projection matrix, while the events and stimulus embeddings are shared across subjects. These features allow for modeling spatial variability across subjects (via different projection matrices) and different temporal alignments (discrete estimation separate for each subject). The stimulus embeddings allow for identifying shared structure.

The model is validated in a simulated experiment and on a dataset of fMRI recordings while subjects listened to computer science lectures. The authors examine recovery of latent states and clusterings of neural activity/stimuli in the simulated data. In the analysis of an fMRI dataset, the authors find variations in sequential activity & spatial coding across subjects. Importantly, they show the learned projections output statistically meaningful clusterings on heldout fMRI runs.

**Strengths:**

The proposed methods are original and significant for the analysis of fMRI datasets during sequential tasks across subjects. The idea to also embed the stimulus makes the learned latent state clusterings more interpretable, and may also help with identifiability. The experimental results relating semantic content of course videos to fMRI recordings across subjects appear very significant and useful for scientific analysis.

**Weaknesses:**

The clarity of the methods and experiment could be improved. One example is it is unclear how the stimulus embedding is learned. At some points, the text implies that the stimulus is treated akin to a subject where a projection matrix is learned for the stimulus in the same way it is for the subjects. However, the stimulus is not included in Algorithm 1. Including more details on how the stimulus is incorporated into the model and learned would help with clarity.

Next, some aspects of the modeling approach appear inconsistent which may limit significance. For example, the events $E$ are defined in voxel space whereas the event segmentation is done in the lower dimensional projection space defined by $W_i$. Next, the model fitting approach appears somewhat ad hoc, and it is not clear it corresponds to a single objective such as maximum likelihood estimation. More justification for the proposed fitting approach and simulated data analysis could rectify these concerns.

**Questions:**

* Can the authors clarify how the stimulus embeddings are learned?

* Why are the events $E$ defined in voxel space when the event segmentation is done in the lower dimensional projection space defined by $W_i$? How does this compare to computing the events $E$ based on projected activity $W_i X_i$? I encourage the authors to consider an alternative fitting method that learns $G$ via the projected neural activity $W_i X_i$.

* How sensitive is the model to initialization / how do results vary across different initializations?

Minor comments - typos
* Figure 2 caption: stimulated -> simulated

**Limitations:**

Yes.

---

> ### Author Rebuttal · Authors · 2023-08-08
>
> 1)
> Yes, the stimulus embeddings were treated like a subject when fitting the model (as you’ve already noted). We include one copy of the stimulus embeddings per each subject in order to prevent the model from over-favoring the human subjects during the forward-backward step. In Algorithm 1, we would adjust the number of subjects, n, to reflect this procedure. Note that because our approach does not require the number of voxels/features to be consistent across subjects/models, the fact that models are in a different feature space does not require any changes to the algorithm.
>
> 2)
> Events at the group level (G) are all defined in the low-dimensional space, and the HMM fitting takes place in this low-dimensional space. The event matrices E (in the original voxel/feature spaces) are computed and used only when updating the projection matrices W. The HMM update procedures, following the expectation-maximization algorithm (Baum-Welch), in combination with PCA updates to the projection matrices, maximizes the likelihood of data under the assumption that voxels have uniform error variance. https://www.sciencedirect.com/science/article/pii/S0169743998000902
>
> 3)
> There is no randomness in initialization - the HMM fitting starts by assuming very high measurement variance, which effectively starts the estimation procedure at the model's prior distribution.

---

> > ### Comment · Reviewer_XnCP · 2023-08-14
> > **Response to comment**
> >
> > Thank you to the authors for their response. Upon consideration my primary concerns are unchanged, as detailed in the comments below.
> >
> > 2) I understand that PCA corresponds to maximum likelihood estimation in a certain generative model, and that EM in an HMM also finds local maxima in the log likelihood. However, it is not necessarily the case that the proposed combination of those steps here ascends a single log likelihood objective. For example, using PCA to determine $G$ implies that $G$ is a latent variable that should be marginalized over to determine the overall marginal likelihood (just as the low-dimensional latent variables from PCA as integrated out to compute the marginal likelihood in PCA). I think connecting the steps of the fitting procedure to an overall objective is important to understand what the algorithm is optimizing, but the connection remains unclear at this point.
> >
> > 3) In algorithm 1 it says $W_i  \leftarrow N_{D \times V_i}(0, 1) \forall i \text{ in } 1...n$. Does that mean the matrices W are initialized as random samples with each element IID Normal with 0 mean and 1 variance? If so, then the initialization is not deterministic, and it is helpful to know how the model and performance vary across runs. If that means something else, I suggest that the authors clarify that line.

---

> > > ### Author Response · Authors · 2023-08-17
> > >
> > > 2. We apologize for our lack of clarity on this point. The overall log-likelihood being optimized is $p(X | W, G)$, i.e. the likelihood of the data given the transform and event pattern parameters. The standard Baum-Welch procedure alternates between an Expectation step in which we estimate the probabilistic latent state assignments for each timepoint (based on the current parameters), and a Maximization step in which we re-calculate the parameters based on the latent state estimates. The Expectation step in our model (computing the η variables in Algorithm 1) is unchanged from the standard procedure, since this takes place entirely within the low-dimensional space with W and G held constant.
> > >
> > >       For the Maximization step, we seek to maximize the η-weighted log probability of our observation model as a function of the parameters W, G. As described in the paper (eqn 1 and the following paragraph), the observation log probability is proportional to the correlation between the projected data and the group-level event patterns, so we seek to maximize $\sum_i \sum_t \sum_e η_{i, t, e} corr(W_i X_{i,t}, G_e)$ which is equivalent, up to a scale factor, to: $\sum_i \sum_e corr(W_i E_{i,e}, G_e)$ This can be thought of as a canonical correlation analysis, in which we seek to find the projections W and G that maximize the correlation between the events E and the Identity matrix. Using the standard solution to CCA, the optimal G is composed of the eigenvectors of the covariance matrix of E (averaged across subjects, since here we seek a shared G that fits all subjects simultaneously). This is equivalent to running PCA on the stacked event representations E as in our model, since each event representation is separately z-scored. Finally, the CCA solution for W is equivalent to linear regression of E onto G - our model performs this step via ridge regression to regularize the matrices W_i (since these have number of columns = number of voxels in a region).
> > >
> > > 3. The reviewer is correct that there is technically a random initialization of the transform matrices, but this description of the Algorithm was misleading on our part. During the very first pass of fitting, the initial event patterns G are all set to the same pattern (the mean of the projected data), and so the projected data plays no role; the state estimates η produced by the forward-backward algorithm are driven solely by the model priors and not the projected data (since each timepoint matches all events equally well). The Ws were randomly initialized only to allow the forward-backward algorithm to run, and are meaningfully set for the first time at the end of the first loop. A more straightforward description of Algorithm 1 would be to say that the fitting process begins with η (set to the model priors), not with G and the Ws.

---

### Official Review · Reviewer_jeCv · 2023-07-04

**Soundness:** 2 fair
**Presentation:** 2 fair
**Contribution:** 2 fair
**Rating:** 4
**Confidence:** 4

**Summary:**

This paper develops Hyper-HMM as a hybrid model that simultaneously aligns both temporal and spatial features across fMRI datasets. The proposed model learns a linearly project that maps voxels to a low dimensional latent space, in which timecourses are segmented into corresponding temporal events. The purpose of this model is to remove the effect of each individual’s mental trajectory through an event sequence, and to also align with other feature spaces like stimulus content. Overall, it is an interesting paper, however, there are several concerns about the machine learning novelty, validating the empirical studies, and the clear presentation of the proposed method.

**Strengths:**

Please refer to the question section


**Weaknesses:**

Please refer to the question section


**Questions:**

The followings are the major concerns and minor comments:


1) It is still unclear to me as to what the novelty of this paper is in terms of machine learning. There may be some contributions to computational neuroscience in this paper; however, which part of the machine learning approach is new in this paper? The author(s) of the proposed paper should specify whether there is anything theoretically new for the proposed approach or if this is merely an application paper for HMM techniques.


2) My second concern is the design of the empirical studies. Even though the study contains several beautiful figures, more numerical analyses would be helpful to convince the reader that the proposed method is effective. There is a lack of regular analysis and conventional machine learning metrics (such as accuracy, dice, etc.) which make it difficult to understand the results.

3)  The proposed method should be benchmarked in comparison with related state-of-the-art techniques.


4) In this paper, the notations are confusing. In regular papers, scalers are denoted by small letters, vectors are defined with small letters (highlighted in bold), and matrices are denoted by capital letters using bold. In this paper, there are a lot of conflicts. It is so hard to trace what is a set, a matrix, or even a distribution.


5) There are some minor linguistic and typo problems in this paper.


**Limitations:**

Please refer to the question section

---

> ### Author Rebuttal · Authors · 2023-08-08
>
> 1)
> Our model incorporates spatial alignment within the Baum-Welch update procedure for the temporal HMM, allowing for simultaneous spatial and temporal alignment. Although the components of this model are indeed taken from previous work, the combined model and its application to fMRI data are novel. We apply our approach to a dataset meant to maximally capture substantial variance across individuals learning and remembering complex pieces of information throughout an extended period of time (an entire semester), which has not been thoroughly examined in cognitive neuroscience/psychological sciences due to the methodological constraints we address in this paper.
>
> 2)
> Unfortunately, our choices in evaluating the model deviated from conventional machine learning metrics. Because of the inherently difficult nature of obtaining “ground truth” labels of an individual’s true cognitive state on a second-by-second basis, metrics from supervised learning are not applicable here. We use conventional machine learning metrics such as R2 to test how well projections generalize across runs.
>
> 3)
> As of yet, there are not any suitable and related state-of-the art techniques against which we can benchmark our current model. Although there are existing spatial alignment, temporal alignment, and brain-stimulus encoding models available, none of these models have explored all three (spatial, temporal, stimulus feature) alignments simultaneously. This prevents us from applying them in cases when both temporal and feature dimensions are not aligned, such as for mapping between fMRI data (timepoints x voxels) and semantic models (sentences x features).
>
> 4)
> Our notation in the main text uses capital letters for matrices (such as the weight matrix, data matrix, etc.) and lowercase letters for scalars, with the exception of the scalar dimensionality D.
>
> 5)
> We have identified a couple typos in the manuscript after the submission deadline; we hope that none of these caused confusion in understanding the paper.

---

> ### Comment · Reviewer_jeCv · 2023-08-20
>
> I have read all the comments and the corresponding responses. I am not satisfied with the responses regarding my concerns and believe that this paper needs an additional revision stage to be ready for the publication process. I will keep my score as it is, however, I am open to other opinions as well.

---

### Official Review · Reviewer_wrdu · 2023-07-07

**Soundness:** 3 good
**Presentation:** 3 good
**Contribution:** 3 good
**Rating:** 6
**Confidence:** 3

**Summary:**

The authors develop a method to identify and align events in the brain and external stimulus. They iteratively fit a Hidden Markov Model to find the times of events, and spatial characteristics of those events.

**Strengths:**

Originality: This work is a minor update to past work, by accounting for time shifting as well.
Quality: The authors use careful validation with simulated data, and careful cross-validation in real data.
Clarity: The paper is largely well written.
Significance: The main problem of comparing neural activity across subjects and referencing those patterns to interpretable stimulus-driven semantics is an important one, especially as the amount of data in the field continues to grow.

**Weaknesses:**

The events in this paper refer to extended time periods. Events are often described as instants in time, rather than extended periods. It would be helpful to carefully articulate the definition of events to prevent misunderstandings.

The authors make quite strong assumptions, despite protestations that their method is very general. In particular, I am skeptical about the "constant events" in the time course, and about the linear embedding of the semantic content.

The authors only test out a narrow range of timing differences (25%), which leads me toward greater skepticism about the generalizality of the author's results.

Suggestion: It would be useful to test this method in a simulation with a deliberately time-warped movie, to check whether the method can recover simulated data can recover the true timercourse.
Minor: Figure 2 caption has a typo: should read "simulated data", not "stimulated data".


**Questions:**

L235: "resulting in latent event representations that are largely orthogonal, which will fail to capture semantically-meaningful relationships and will fail to generalize to new events.” I don’t understand. Orthogonal components in event relationships should be common and natural. Why is orthogonality considered as meritorious?

Three dimensions is shockingly low dimensionality of semantic space. This seems to me like analyzing image data and find the dominant principal component, the sky.

Linear projections onto semantic space is highly restrictive and unrealistic. But it could be a reasonable lower bound on alignment.

**Limitations:**

The authors do an excellent job of articulating their problem and the essence of their solution..

---

> ### Author Rebuttal · Authors · 2023-08-08
>
> 1)
> Here we rely on a definition of events, and event segmentation, widely used in psychology and cognitive neuroscience. Continuous streams of information, as is the case in videos or text, can be divided into smaller and smaller chunks (e.g. a book consists of chapters, chapters are composed of paragraphs, paragraphs of sentences, sentences of words, so on and so forth). Psychologists use this definition to study the ways in which people organize meaningful chunks of information (Yates, Sherman, & Yousif, 2023; Zacks et al., 2007), and have found that this may be the foundation in remembering contextually- or semantically-linked information (Ezzyat & Davachi, 2011). With respect to temporally-constrained information, such as the class videos we use in our paper, an event consisting of semantically related adjacent timepoints is an extended period of time.
>
> 2)
> Our simplifying assumptions are driven by the limitations of fMRI datasets, which have high levels of noise and generally consist of only ~1000 timepoint samples per subject. The “constant events'' we refer to in our paper depend on event stability, or a spatial (voxel) pattern in the brain that appears across adjacent time points. This pattern stability is viewed as the neural instantiation of a stable event representation (Antony, 2021; Baldassano et al., 2017; 2018). Brain representations are of course never fully static, but we can define events at the shortest timescale for which there are meaningful dynamics that generalize across subjects.The use of linear embeddings of the semantic content is a form of learning linear mappings from neuroimaging data to semantic spaces, a standard practice in psychology/cognitive neuroscience, such as when creating encoding models of the brain (Naselaris, 2011).
>
>
> 3)
> The simulations were intended only to show proof-of-concept behavior for the model, demonstrating that the architecture and fitting procedure is able to capture varying temporal onset/offset of events and topographical differences across individuals while applying increasingly high spatial noise. Overall, simulating realistic fMRI data is an ongoing challenge, given the highly complex spatial and temporal correlations present in a real dataset. We hope to have access to simulated fMRI datasets that are more representative of real data sometime in the future.
>
> Q1) Orthogonality across events would be indicative of events being equally distinct from each other, thus producing no meaningful event similarity structure shared across subjects. We believe that one strength of this model is the ability to learn a shared event space while preserving individual nuances. This aspect would be lost if the model were tuned to individuals without any regard to the shared event space.
>
> Q2) Given that the model is fit to data from single regions at a time, the low number of dimensions is consistent with pre-existing literature, such as seen with a commonly accepted fMRI dimensionality reduction algorithm, the Shared Response Model (SRM) <https://papers.nips.cc/paper_files/paper/2015/file/b3967a0e938dc2a6340e258630febd5a-Paper.pdf>.
>
> Q3) While linear projections onto a semantic space can be limiting at times, fMRI data itself is limited per subject. Our training and test datasets contained roughly 1000 timepoints on average, requiring the use of low-complexity models to accommodate the nature of the dataset.

---

> > ### Comment · Area_Chair_2NSY · 2023-08-22
> > **Response to rebuttal**
> >
> > Dear Authors,
> >
> > Thank you for answering the points that were raised, your response will be taken into account.
> >
> > Best,
> >
> > Your Area Chair

---

### Official Review · Reviewer_jAF3 · 2023-07-17

**Soundness:** 3 good
**Presentation:** 3 good
**Contribution:** 2 fair
**Rating:** 6
**Confidence:** 4

**Summary:**

UPDATE:

I have raised my score and now support acceptance of this paper. I believe it will be a good contribution to the conference.

-----------------------------
The authors propose an extension for an HMM model proposed by Baldassano and colleagues to align interindividually different brain responses in both space and time to analyze naturalistic stimulation settings. They test their approach through simulations and on an empirical dataset of 19 participants, who watched a series of 5 computer science lectures. They show that their model is robust to a number of noise levels in the simulation analysis. In the empirical analysis, the model appears to learn a meaningful latent space (concrete vs abstract and future-oriented vs present tense descriptions) and outperforms a null model.

**Strengths:**

- The research problem addressed is an interesting and timely problem, namely how to compare spatially and temporally heterogeneous responses across participants
- The paper is largely well-written


**Weaknesses:**

- The sample size is small
- The approach is not compared to other state-of-the-art models, rather the authors compare it to a null model, which is poorly explained
- The authors do not provide analysis code or sufficient detail to reproduce their results

**Questions:**

Overall, this work tackles an interesting problem and shows some first promising results, but I believe that it would require quite some work like more extensive simulations, comparisons to other methods and testing in a larger dataset or a second dataset (to demonstrate that this method is not only useful for applications to naturalistic stimulus analysis, but other areas of neuroscience research, for example aligning multiple imaging modalities) to meet the NeurIPS standard.

Nonetheless, I really enjoyed reading the paper and I strongly encourage the authors to continue this interesting work. Here are some questions/comments that would allow me to increase my score. I hope you find them helpful and constructive.

**Major points**
- The actual model could be explained in greater detail. It would be helpful to include pseudo-code for the algorithm and the cost function explicitly in the manuscript.
- Please, explain (at least in the supplement) the choice of preprocessing, e.g. quality control (motion detection/scrubbing, motion artifact rejection thresholds), software versions, scanner/coil details, filtering, slice-time alignment if conducted etc.
- The choice of the simulation parameters is not clearly motivated, can you explain why you chose to simulate datasets with only 6 participants or only 5 voxels, this does not correspond well to empirical situations, where the sample would be larger, and #voxels >> #stimuli, which would be important to test whether your reweighting (p.4 125-126) works. It would also be helpful to express the noise as SNR to assess whether your method assumes realistic noise.
- For the empirical analysis, it is unclear how you construct your baseline. Please, be more specific about how you computed that, otherwise it is impossible to assess the model performance
- Related to that Figure 5 is not very clear, I am assuming that the black dot and the black diamond in Fig 5 reflect the actual value, whereas the grey shaded area is the null-distribution? This should be clearly stated in the figure captions.
- Figure 5: It is quite striking that the fMRI stimulus match is much worse on the empirical data than the clustering. Based on the simulations (Figure 2), this is unexpected. Could you elaborate on what the reasons for this could be?
- Figure 5: The fMRI-stimulus match null-distribution(?) is centered on a negative R^2. Can you explain why this happens? Could this be suggesting that the baseline model is not appropriate?
- Will code be provided to ensure reproducibility of the analyses?
- p. 6 ll. 218-221: You state that the latent dimensions appear to map onto semantically meaningful dimensions (concrete vs abstract and future-oriented vs present tense descriptions). Is this in line with the literature? Further discussion of this would be helpful.
- p. 9 l. 268-270: “Our experiments demonstrate that [...] this model […] can learn meaningful mappings” To show that these mappings are meaningful, it would be useful to show that the latent scores correlate with an external measure that was not shown to the model, but would be hypothesized to relate to the captured dimensions (e.g., IQ, exam scores, abstract thinking scores or something of that sort). Is any information like this available. That would strengthen your claims.


**Minor points**
- Could you elaborate on the relationship between your method and representational similarity analysis (https://doi.org/10.3389/neuro.06.004.2008)? You very briefly touch on related work (e.g. from Haxby), but I think this could be expanded a bit.


**Limitations:**

The authors have mentioned some limitations (linear assumptions), but the small sample size is not discussed.

---

> ### Author Rebuttal · Authors · 2023-08-08
>
> 1)
> We provided an algorithm (pseudocode) in the appendix submitted as part of our supplementary materials.
>
> 2)
> We used published data provided by Meshulam et al. here: https://openneuro.org/datasets/ds003233/versions/1.2.0 - please see the original dataset for details about the fMRI acquisition. We included a preprocessing script and directions for executing fMRIprep in the readme file in the supplementary material, and did not perform any additional preprocessing (e.g. slice-time alignment) beyond this.
>
>
> 3)
> The simulations were intended only to show proof-of-concept behavior for the model, demonstrating that the architecture and fitting procedure is able to capture varying temporal onset/offset of events and topographical differences across individuals while applying increasingly high spatial noise.Overall, simulating realistic fMRI data is an ongoing challenge, given the highly complex spatial and temporal correlations present in a real dataset. We hope to have access to simulated fMRI datasets that are more representative of real data sometime in the future.
>
>
> 4)
> Our baseline was to use random projections for each subject/model into the latent space, with each element of the project matrix sampled from a standard normal distribution N(0, 1). This approach preserves the temporal structure for each subject (since this projection is held constant within subjects), while disrupting alignment between different subjects.
>
>
> 5)
> Yes, that is the correct interpretation of Figure 5. We only briefly mention the null distribution in the last line of the captions and should have indicated that they are represented by the gray shaded area.
>
> 6)
> Aligning fMRI data to the stimulus in the real dataset is a much more challenging task than fMRI-fMRI alignment. Rather than relating the same kind of data across people – which tends to have some similarities even before applying spatial or temporal alignment – this is an entirely different model with stimulus features that are only partially related to fMRI responses. This complexity is not straightforward to capture in our models, though note that at higher noise levels we do see performance that is reduced and more variable for the stimulus alignment (since there is only a single stimulus to align, as opposed to multiple subjects).
>
> 7)
> We would obtain R2 = 0 if the stimulus projections, on average, sit at the mean of the fMRI data. This means that random projections are likely to be below 0 (worse) since the stimulus projections will be randomly distributed with respect to the fMRI data. Because the fMRI data and stimulus model come from different representational spaces with differing dimensionalities, there is no simple baseline alignment between the two spaces.
>
> 8)
> We provided all code necessary to reproduce the entire pipeline from preprocessing the publicly available dataset (please see above response for preprocessing fMRI data) all the way to fitting the model in the supplementary material.
>
> 9)
> These axes of semantic variation have been observed in prior fMRI studies. Gilead et al., 2013 (DOI: 10.1016/j.neuroimage.2012.09.073) show differences in activity for concrete vs abstract and future- vs past- and present- tense sentences, and concrete vs abstract concepts engage different regions of the semantic network (Conca et al., 2021; DOI: 10.1038/s41598-021-02013-8).
>
>
> 10)
> Mapping the trajectories in the latent space to a behavioral measure would be a great validation of the method, but this would require a dynamic behavioral measure of second-by-second cognitive states; no such metric exists for this dataset, and it is also unclear how we could obtain this kind of measure without disrupting a student's comprehension of the material. Instead, we are arguing that our latent mappings are meaningful in the sense that they can generalize across runs (i.e. successfully align new fMRI-stimulus data).
>
>
> Minor 1) Similar to RSA, we make the assumption that event representations are linearly related across subjects and share the same similarity structure. However, RSA does not identify any temporal or spatial alignment, unlike work from Baldassano et al. or Haxby et al., respectively.

---

> > ### Comment · Reviewer_jAF3 · 2023-08-14
> > **Most points addressed, except simulations**
> >
> > Thank you very much for addressing most of my concerns. There are only two points, one major and one minor remaining:
> >
> > Major:
> > 1) The simulation is still a major concern: Since the simulation is supposed to test whether the method works in principle, I think the simulation settings are paramount. This is especially the case in your study, since the empirical data is very small. I would strongly suggest to include simulations with more appropriate settings (larger number of voxels or ROIs, exploring the relationship with temporal and spatial noise). I think reviewer 1rZM made some helpful suggestions in terms of how to go about that. Resting state data that could be added as noise, can be freely downloaded from large repositories.  Alternatively, you could use other models such as the Wong-Wang-Deco model (https://doi.org/10.1523/JNEUROSCI.5068-13.2014) to simulate resting-state and add task-based perturbations, or use dynamic causal models (https://doi.org/10.1016/S1053-8119(03)00202-7) to simulate task based data. If you conduct more realistic simulations, I would be willing to raise my score.
> >
> > Minor:
> > 2) I suggest to include the small sample size in the limitation section or do you plan to include a test on another second data set?

---

> > > ### Author Response · Authors · 2023-08-17
> > >
> > > Major:
> > > 1. We have posted results and an explanation for an additional simulation experiment.
> > >
> > >
> > > Minor:
> > >
> > > 2. We have additionally applied the model to data collected during a movie-watching paradigm (from Aly et al., DOI: 10.1162/jocn_a_01308) and observed similar performance for aligning fMRI events across subjects. We can include these results in the final version of the paper as additional validation.

---

> > > > ### Comment · Reviewer_jAF3 · 2023-08-17
> > > > **Thank you very much and all concerns addressed**
> > > >
> > > > Thank you very much. I very much appreciate the hard work the authors put into this to run these additional analyses. I believe that these additional analyses have greatly strengthened the paper. I would also recommend including a brief sentence about the SNR breakpoint for the method in the limitation section as a cautionary note for colleagues who seek to apply this method.
> > > >
> > > > Thank you very much again. I will raise my score in light of these revisions. I believe this will be a great contribution to the conference and the field of neuroscience!

---

### Official Review · Reviewer_1rZM · 2023-07-17

**Soundness:** 4 excellent
**Presentation:** 4 excellent
**Contribution:** 4 excellent
**Rating:** 8
**Confidence:** 4

**Summary:**

In natural tasks that share input stimuli across participants, the cognitive states or neural responses of different participants might undergo approximately synchronous but slightly jittered dynamics. At the same time, the distribution of neural signals are not exactly consistent across participants at voxels with the same spatial coordinates. These two problems were addressed by two approaches separately by event segmentation and functional alignment. This paper proposes a new approach H-HMM that combines the advantage of both methods. It performed simulation to evaluate its performance and tested it on an fMRI datasets of college students watching the same series of lectures in computer science. The performance appears impressive and achieves what the model is designed for. Further, the paper also demonstrated alignment between fMRI data and semantic features of contents in the lecture.



**Strengths:**

* Simultaneous achieving temporal and spatial alignment was not done before.
* Comprehensive evaluation of the method is performed on both simulated and real data and showing good performance.
* The illustration is generally clear and easy to understand.
* The approach holds promise for a wide range of application and should be a significant contribution to the field

**Weaknesses:**

* I think there is a mismatch between the data size of simulated data and that of fMRI data or semantic features, making it a bit difficult to evaluate the expected performance of this method in general. The simulated data have only 5 voxels/features, which is rarely observed in the domain where this algorithm should be applied. The described numbers also do not match up: 4 events with 9-12 time points per event give rise to fewer than 48 time points, yet it is said that the simulated data have 60 time points. It is also strange that the event is represented by binary patterns. Maybe this is just for the purpose of easy visualization but I worry that the performance might depend on these unrealistic properties in the simulation. Doing it with similar property as would be expected for the fMRI data should be more convincing. If you encounter issue that the actual fMRI data have lower effective dimensionality due to smoothness, it may be simulated by spatial Gaussian process or other ways but I don't think it is justifiable to start with low-dimensional data, as you did not perform PCA on fMRI data to get similar dimensionality before application of the algorithm.
* The description for the update of group-level G, starting from line 122, is a bit confusing. After the stacking, what is the dimensionality of the matrix? I assume the stacked data have a total number of elements as the number of events * (total number of voxels+ semantic features) * D (since E is in the D-dimensional space). If your stacked data have two dimensions (which I assume is the case since you can do PCA on it), which of the two dimensions (after explaining their size) do you treat as data features and which do you treat as samples in the definition of PCA? After projecting the stacked data for each event into D-dimensional PC space, do you need to do anything you do to get G (which is of the shape of number of events * D)?

**Questions:**

My major questions are asked in the weakness, which I think are addressable.

**Limitations:**

Yes.

---

> ### Author Rebuttal · Authors · 2023-08-08
>
> 1)
> The simulations were intended only to show proof-of-concept behavior for the model, demonstrating that the architecture and fitting procedure is able to capture varying temporal onset/offset of events and topographical differences across individuals while applying increasingly high spatial noise.Overall, simulating realistic fMRI data is an ongoing challenge, given the highly complex spatial and temporal correlations present in a real dataset. We hope to have access to simulated fMRI datasets that are more representative of real data sometime in the future.
>
> The 4 events were randomly selected in each individual to have anywhere from 12 - 18 time points per event. The range reported in the submission was from an earlier round of simulations and mistakenly used instead of the most recent parameters for the simulations. We aimed to ensure that each event would have high variance across individuals in terms of onset and offset times.
>
> 2)
> The voxel/feature vectors for each subject/model are concatenated together for each event, yielding a two-dimensional matrix with rows = events and columns = all concatenated voxels/features. After PCA, this long catenated dimension is reduced, yielding a matrix G where the dimensions are events x D (latent dimensionality). The values in the latent dimensions are the data features corresponding to each respective event.

---

> > ### Comment · Reviewer_1rZM · 2023-08-12
> >
> > Thanks a lot fore replying to my comments.
> > I still like the paper.
> >
> > But I cannot agree with the difficulty of simulation:
> > You can still simulate according to your model but simply increasing the number of voxels to those commonly observed, and generate noise with spatial-temporal Gaussian Process.
> > Also, although not perfect, there are R-based fMRI simulator called neuRosim and a python-based one in BrainIAK.
> > Lastly, one can also add pattern that mimics the spatial-temporal smoothness to resting-state fMRI data (but keeping the HMM and sequential order of patterns) and use resting-state data as noise.
> > All of these will be more realistic than the simulated data used here.

---

> > > ### Author Response · Authors · 2023-08-17
> > >
> > > Thank you for your feedback. We understand your concerns and have posted results and an explanation for an additional simulation experiment.

---

> > > > ### Comment · Reviewer_1rZM · 2023-08-17
> > > >
> > > > Thanks a lot for the effort! This new simulation result resolved my concern, so I am happy to increase my score.

---

### Author Response · Authors · 2023-08-17
**Additional simulation experiments**

We have now conducted a simulation experiment with more realistic parameters, taking into account the spatial and temporal characteristics of fMRI data. We used the fmrisim utility in the BrainIAK toolbox to generate a pure-noise dataset in the Angular Gyrus (using the default fMRI noise parameters, from the fmrisim tutorial) that is same size as our real fMRI dataset (19 simulated subjects, each with 21 runs of the same duration as in the real dataset). We simulated an event-structured signal by assigning each timepoint to an event, with varying timing across subjects (using the same timing parameters as in the original simulation). Within each event there was a constant event pattern signal; these patterns came from a subject's real fMRI data, averaged in 15-TR increments. By combining the signal and noise together with varying weights, we can produce simulated datasets with the same complexity as a real dataset, but for which we know the ground-truth timing and patterns within each subject.

We measured the performance of the H-HMM in recovering both spatial projection and temporal alignment. For spatial projection, we used the same procedure we applied to the real fMRI dataset in the paper: we fit the H-HMM to half of the runs in the simulated dataset, and then measured the extent to which the learned projection matrices could be used to align the events in the held-out runs using Variance Explained. For temporal alignment, we fit the H-HMM to the full simulated dataset and compared the event boundaries identified for each subject to the ground-truth boundaries. We computed the fraction of ground-truth boundaries that were correctly recovered (within one timepoint); for comparison, a null model of randomly-placed boundaries achieves 19.0% on this measure. Note that even with high SNR, both of these tasks are non-trivial, since neighboring event patterns can be highly similar and the model must map Angular Gyrus event patterns into a compressed latent space with D=3.

Results:
*  Signal:Noise | Projection VE | Boundary recovery |
*  2.0:1 | 95.8% | 63.8% |
*  1:1.0 | 93.6% | 62.5% |
*  1:1.5 | 89.7% | 62.1% |
*  1:2.0 | 82.2% | 54.4% |
*  1:2.5 | 74.6% | 48.5% |
*  1:3.0 | 58.8% | 45.2% |
*  1:3.5 | 49.4% | 34.7% |
*  1:4.0 | 47.0% | 24.8% |

For high SNR, the model is highly accurate at identifying the spatial transform between simulated subjects and can successfully identify the majority of the event transitions. Performance on both measures is degraded for SNR < 1:3.5, with spatial alignment performance dropping below our observed performance on real data and boundary identification that approaches chance. We hope that this new analysis helps to further support our claims that the method is robust to realistic dataset sizes and noise properties; we will replace our simulation analysis with this more sophisticated version in the final version of the paper.

---

### Decision · Program_Chairs · 2023-09-21

**Decision:**

Accept (poster)

**Comment:**

This paper introduces an interesting method for aligning cognitive events across individuals using their brain recordings and stimulus features. I believe the authors have provided sufficient response to the reviewer's concerns, including running a realistic simulation. Some concerns were raised about the machine learning novelty in the paper, however, neurips has interest in machine learning methods applied in novel ways to answer neuroscientific problems, so this paper is appropriate for neurips. The authors are requested to address all the concerns raised in the review period and to add their new results and explanations.